# The Endocannabinoid/Endovanilloid System in Bone: From Osteoporosis to Osteosarcoma

**DOI:** 10.3390/ijms20081919

**Published:** 2019-04-18

**Authors:** Francesca Rossi, Chiara Tortora, Francesca Punzo, Giulia Bellini, Maura Argenziano, Alessandra Di Paola, Marco Torella, Silverio Perrotta

**Affiliations:** 1Department of Women, Child, and General and Specialized Surgery, University of Campania Luigi Vanvitelli, 80138 Naples, Italy; chiara.tortora@unicampania.it (C.T.); punzofrancesca.phd@gmail.com (F.P.); maura.argenziano@unicampania.it (M.A.); alessandra.dipaola@unicampania.it (A.D.P.); silverio.perrotta@unicampania.it (S.P.); 2Department of Experimental Medicine, University of Campania Luigi Vanvitelli, 80138 Naples, Italy; giuliabellini@hotmail.com; 3Department of Gynecology, Obstetrics and Reproductive Sciences, University of Campania Luigi Vanvitelli, 80138 Naples, Italy; marco.torella@unicampania.it

**Keywords:** CB1, CB2, TRPV1, bone, osteoporosis, osteosarcoma, osteoclasts, osteoblasts

## Abstract

Bone is a dynamic tissue, whose homeostasis is maintained by a fine balance between osteoclast (OC) and osteoblast (OB) activity. The endocannabinoid/endovanilloid (EC/EV) system’s receptors are the cannabinoid receptor type 1 (CB1), the cannabinoid receptor type 2 (CB2), and the transient receptor potential cation channel subfamily V member 1 (TRPV1). Their stimulation modulates bone formation and bone resorption. Bone diseases are very common worldwide. Osteoporosis is the principal cause of bone loss and it can be caused by several factors such as postmenopausal estrogen decrease, glucocorticoid (GC) treatments, iron overload, and chemotherapies. Studies have demonstrated that CB1 and TRPV1 stimulation exerts osteoclastogenic effects, whereas CB2 stimulation has an anti-osteoclastogenic role. Moreover, the EC/EV system has been demonstrated to have a role in cancer, favoring apoptosis and inhibiting cell proliferation. In particular, in bone cancer, the modulation of the EC/EV system not only reduces cell growth and enhances apoptosis but it also reduces cell invasion and bone pain in mouse models. Therefore, EC/EV receptors may be a useful pharmacological target in the prevention and treatment of bone diseases. More studies to better investigate the biochemical mechanisms underlining the EC/EV system effects in bone are needed, but the synthesis of hybrid molecules, targeting these receptors and capable of oppositely regulating bone homeostasis, seems to be a promising and encouraging prospective in bone disease management.

## 1. Endocannabinoid/Endovanilloid (EC/EV) System in Bone

Despite its rigidity, bone can be defined as an extremely dynamic organ, in a constant remodeling process by osteoblasts (OBs) and osteoclasts (OCs), key cells in maintaining bone homeostasis and in answering to mechanical stresses [1,2]. Osteoclasts secrete acid and proteinases for cartilage resorption, whereas osteoblasts replace new bone and also influence the osteoclast activity. Osteoblasts express the receptor activator of nuclear factor Kappa-B ligand (RANK-L) which enhances osteoclastogenesis by binding its specific receptor (RANK) onto the osteoclast precursors’ surface. The binding between RANK receptor and its ligand leads to the recruitment of molecules, such as TNF receptor-associated factor 6 (TRAF6), with the consequent activation of MAPK cascade, NF-kB, AKT/PKB, JNK, ERK downstream signaling pathways, and the final expression of genes involved in osteoclastogenesis [3]. Osteoblasts also release osteoprotegrin (OPG), a soluble glycoprotein that acts as a “decoy” receptor binding to RANK-L and thus inhibiting osteoclast activation [4]. In 2016, Luo et al. identified a new receptor for RANK-L, LGR4, that binds RANK-L and suppresses RANK signaling [5]. Considering its crucial role in bone homeostasis, any deregulation in RANK/RANK-L signaling leads to pathological processes such as postmenopausal osteoporosis or cancer-induced bone destruction [6,7,8]. 

The endocannabinoid (EC) system is composed of endogenous cannabinoid ligands (anandamide (AEA) and 2-arachidonoylglycerol (2-AG)), their specific receptors, the cannabinoid receptor type 1 (CB1) and type 2 (CB2), and all the enzymes involved in their synthesis and degradation [9]. The endocannabinoids physiologically interact with several kinds of receptors, one of them being the transient receptor potential cation channel subfamily V member 1 (TRPV1) [10], a subfamily of Ca^2+^ permeable channel which constitutes the endovanilloid (EV) system. The EC/EV system is proven to be involved in the regulation of several physiological processes, such as appetite control, energy balance [11], pain perception [12], and immune response [13]. Moreover, it has been proposed as an anticancer target by several studies [14,15]. The CB1 receptor is more expressed in the central nervous system (CNS), whereas the CB2 receptor can be found predominantly in peripheral tissues [16], even though there is growing evidences indicating that it is also present in the brain [17,18]. In 2017, Liu et al. detected the CB2 receptor in various brain regions in a murine model with immunohistochemistry and in situ hybridization [19]. TRPV1 is a nociceptor predominantly expressed on sensory nerve fibers of the somatic and autonomic afferent neurons [20]. 

Bone cells express CB1 and CB2 receptors and TRPV1 channels and locally produce the endocannabinoids AEA and 2-AG and the enzymes involved in their synthesis and degradation [21,22,23,24]. The pharmacological modulation of these receptors contributes to the maintenance of bone mass by stimulating stromal cells and osteoblasts and by inhibiting monocytes and osteoclasts [25,26] (Figure 1). In particular, the inactivation of CB1 by its inverse agonist, AM251, or its genetic deletion, inhibits OBs’ differentiation from the bone marrow-derived cells, reducing a specific osteoblast transcription factor, runt-related transcription factor 2 (RUNX2) [27]. In addition, TRPV1 stimulation by its selective agonist, resiniferatoxin (RTX), reduces RUNX2, OPG, and alkaline phosphatase (ALP), inhibiting calcium deposition by osteoblasts. Conversely, CB2 stimulation by its selective agonist, JWH-133, inhibits the release of RANK-L, consequently reducing osteoclasts’ number and differentiation and leading to mineral deposition [23]. CB2 receptor stimulation acts as inductor of bone matrix deposition, whereas the TRPV1 receptor stimulation acts as inhibitor of osteogenic signaling. Moreover, RTX, in in vitro osteoclasts, is able to increase the expression and the activity of two important osteoclast biomarkers, tartrate-resistant acid phosphatase (TRAP) and cathepsin K. In particular, the very first evidence of TRPV1 expression in human osteoclasts was formulated in 2009, when the co-expression of TRPV1 and cannabinoid receptors in these cells was demonstrated [22]. While their co-expression in neuronal [28,29] and non-neuronal cells, such as HEK cells and mice osteoclasts and osteoblasts [10,30], had already been proven, it has never been proven before in human bone primary cells. The co-expression of these receptors and the fact that they can be stimulated by the same ligands give evidence of a cross-talk between EC and EV receptors. This interaction, especially between CB2 and TRPV1, is also evident in bone tissue where they oppositely modulate osteoblast and osteoclast activity [23,31].

The important evidence that cannabinoid and vanilloid receptors are co-expressed in mouse and human bone suggests that they might act together to balance the bone mineralization and resorption by different actions of AEA on TRPV1 and cannabinoid receptors. 

On this basis, pharmacological modulation of the EC/EV system may be a valid approach in the prevention and treatment of bone diseases.

## 2. Endocannabinoid/Endovanilloid System in Osteoporosis

The EV/EC system plays a pivotal role in regulating bone cell activity even though the conclusions in the literature about its role have been discordant for long time. CB1 and TRPV1 stimulation activates osteoclasts [21,31,32,33,34,35], while CB2 represents the counterpart for bone mineralization and remodeling via osteoclast inhibition [26,36] (Figure 2). In 2005, Idris et al. observed that selective agonists at CB2 receptor (HU308 and JWH-133) induce RANK-L-mediated osteoclast formation [35], while in 2008, Bab et al. demonstrated that CB2-deficient mice have a normal phenotype at birth, but they undergo a bone mass reduction over time, suggesting a progressive age-related bone loss [37]. Since then, several papers have reported that the activation of CB2 enhances bone formation and limits bone resorption [24]. For example, a CB2-selective agonist (HU308) causes the inhibition of RANK-L-induced osteoclast formation in RAW 264.7 culture in vitro [33]. 

In 2005, Karsak et al. demonstrated that a polymorphism in the CNR2 gene, which encodes the CB2 receptor, is strongly associated with bone mineral density in a population of postmenopausal OP patients [38]. This polymorphism, a missense variant (Gln63Arg), that affects CNR2 expression and activity is significantly associated with the most severe form of OP [38,39]. Thus, a reduced expression or efficacy of CB2 signaling results in a lower bone density and even osteoporosis (OP). OP represents the most common metabolic bone disease with the highest impact on public health with relative costs [40,41]. It is characterized by decreased bone mineral density, reduced bone strength, and consequent increase in skeletal fragility and the occurrence of fractures [42,43]. A dysregulation of bone homeostasis with a decreased activity of osteoblasts and osteocytes and an enhanced osteoclast activity (Figure 3A,B), generally, characterize OP [44]. In postmenopausal OP, the most frequent type of primary OP, the gonad function interruption seems to retard osteoclastic but not osteoblastic apoptosis, inducing a progressive bone mass loss that, surprisingly, does not benefit from hormonal replacement treatment [45,46,47]. To completely restore the bone balance in OP, a successful therapy should increase osteoblast activity and at the same time decrease the osteoclast activity. However, many available therapies affect only the catabolic osteoclast activity [48]. Postmenopausal OP is the most common metabolic bone disease, but secondary factors such as drugs, iron overload, and pathological conditions may significantly increase the risk of bone loss and skeletal fragility [49].

Accordingly, it has been demonstrated that CB2 receptor is differently expressed in osteoclasts from menopausal women with or without OP and that the CB2 stimulation inhibits activity and differentiation of osteoclasts from osteoporotic postmenopausal women [50]. In post-menopause, estrogen withdrawal seems to delay osteoclast apoptosis inducing a lower bone mass density [47,51,52]. Several studies report that estrogens are able to modulate cannabinoid receptor expression both in rats and in humans [53,54,55]. Interestingly, the 17-β-estradiol induces an increase in CB2 expression through the recruitment of a putative estrogen responsive element in the CB2 encoding for gene, suggesting the possibility of acting on this receptor instead of adopting a hormonal therapy to reduce bone resorption in postmenopausal OP [55]. CB2 receptor acts also on osteoblasts and their precursors, promoting bone formation [26,56]. Scutt and Williamson demonstrated that the administration of cannabinoids in vivo stimulates the recruitment of MSCs from the bone marrow through CB2 receptor activation [56]. Ofek et al. revealed that the bone marrow stromal cells derived from CB2-deficient mice lack bone deposition under osteogenic stimuli [26]. According to this evidence, it has been demonstrated, for the first time in 2015, that CB2 receptor is expressed on human osteoblasts differentiated from bone marrows of healthy donors and that CB2 stimulation increases human osteoblast activity [23]. These studies prove that peripheral CB2 receptors have a protective role in bone metabolism and, therefore, targeting these receptors may represent a novel approach for the treatment of bone-related disorders, such as postmenopausal OP.

Furthermore, CB1 receptor is known to modulate both osteoblast and osteoclast activity [35,57,58]. In detail, CB1 exerts age-dependent effects on bone mass by regulating the differentiation of osteoclasts and MSCs into osteoblasts and adipocytes. Idris et al. were the first ones to report that genetic inactivation of CB1 receptor results in higher bone mass in young mice [35,57,58]. In particular, mice with CB1 deficiency present increased bone mass already at three months of age, but develop age-related OP associated with the accumulation of adipocytes in bone marrow [35]. Accordingly, Gimble et al. demonstrated that MSCs from elderly subjects have a reduced capacity to differentiate into osteoblasts and an increased capacity to differentiate into adipocytes, which implies the accumulation of fat in the bone marrow with aging [59]. Therefore, CB1 ligands may be used to enhance bone mass and prevent age-related OP. 

Moreover, several in vitro and in vivo studies [35,60] have demonstrated the direct effects of CB1 on osteoclast differentiation. In particular, Samir and Malek demonstrated a reduction of RANK-L gene expression and an increase in OPG expression in young rats upon CB1 inhibition [60], and Idris et al. suggested that the defective osteoclast differentiation in CB1-deficient mice is caused by a reduced RANK-L expression which impairs the ability of osteoblast to support the osteoclast differentiation [35]. A similar result has been observed in vitro using inverse agonists (AM251 and SR141716A) at CB1 that cause osteoclast apoptosis and a reduction in their differentiation [33,36]. Therefore, CB1 and CB2 have distinct roles in bone homeostasis and their individual blockage may be harmful, however their combined inhibition may be beneficial in preventing age-related bone loss [61]. 

TRPV1 is another potential target for preventing OP: TRPV1-/- mice in fact have higher bone mass density (BMI) than wild-type (WT) animals [62,63]. The underlying reasons are still unclear, but in 2017, He et al. gave a first evidence in TRPV1 -/- mice that the osteoclast precursors present a reduction in calcium levels after stimulation with RANK-L; specifically, they poorly answered to osteoclastogenic stimulus [32]. Indeed, RANK-L signaling leads to Ca^2+^ oscillations that are responsible for a Ca^2+^/calcineurin-dependent osteoclast differentiation. According to these studies, the TRPV1 genetic ablation or pharmacological inhibition protects against ovariectomy-induced bone loss by affecting osteoclast activity. Moreover, TRPV1 pharmacological inhibition with agents like capsazepine prevents bone loss, thereby enhancing osteoblast differentiation. Taken together, the evidence from these studies suggests that TRPV1 deletion or inhibition could influence the bone remodeling process, by affecting osteoclast and osteoblast differentiation and in particular promoting new bone formation. Based on all this evidence, an enhanced activation of TRPV1 and CB1 may be responsible for osteoclast activation and for bone resorption in osteoporosis. Accordingly, in osteoclasts from osteoporotic patients, TRPV1 channels are upregulated, differently localized, and rapidly responsive to activation and, in turn, to desensitization. In effect, in these cells, the stimulation of the channel with the agonist RTX causes immediate activation and desensitization leading to the same effect of TRPV1 antagonism. Moreover, TRPV1 stimulation induces an overexpression of CB2 receptors providing evidence of a functional cross-talk between CB2 and TRPV1 receptors in OP [50]. 

The EV/EC system is also dysregulated in a secondary form of OP, such as the glucocorticoid-induced OP. Wang et al. demonstrated that bone tissue responds to supra-physiologic levels of glucocorticoids (GCs) with decreased bone formation [64,65], and Samir et al. and McLaughlin et al. reported that glucocorticoids promote osteoclastogenesis by increasing RANK-L and decreasing OPG expression [60,66]. 

The glucocorticoid-induced OP is the most common form of secondary OP [67]. Glucocorticoids, widely used for inflammatory and autoimmune disease treatment, affect both directly and indirectly the bone cell activity leading to a bone mass reduction that occurs, independently of sex and age, in 30–50% of treated patients [64,65,66]. Several studies have demonstrated that a chronic glucocorticoid therapy is strongly associated with a low bone mineral density (BMD) and a high susceptibility of fractures [68,69,70,71]. Sosa et al. and Migliaccio et al. demonstrated that prednisolone treatment induces apoptosis of osteoblasts and osteocytes that leads to the reduction of bone formation [70,71]. 

Modulating the EC/EV system can limit GC side effects on BMD. When they are co-administered with CB1 antagonists, they induce an improvement of BMD in young rats, while leading to its decrease in old ones, thus confirming the role of CB1 receptor antagonists in age-related bone turnover [60]. These studies suggest that CB1 antagonist can be used to prevent glucocorticoid-induced OP in youth but should be avoided in old age. It has also been demonstrated that methylprednisolone inhibits CB2 and increases TRPV1 signaling in human osteoclasts, suggesting that pharmacological compounds stimulating CB2 or inhibiting TRPV1 might reduce, probably inhibiting protein kinase C beta II, also the methylprednisolone-induced osteoclast over-activation [62,72]. 

Studies have reported that iron is also an important risk factor for OP [73,74]. Iron overload is generally a consequence of chronic blood transfusions that are necessary in disorders such as beta thalassemia major (TM), hereditary hemochromatosis, and sickle cell anemia [75]. In vivo and in vitro studies suggest that iron excess directly influences bone formation and remodeling [73,76].

Tsay et al. suggested the importance of iron-induced ROS on bone metabolism, demonstrating that iron overload in mice results in increased bone resorption and oxidative stress, leading to changes in bone microarchitecture and bone loss [73]. Balogh et al. demonstrated, both in vitro and in vivo, that iron is able to directly inhibit osteogenic commitment and bone marrow stromal cell (BMSC) differentiation [76]. However, the mechanism through which iron induces bone loss is still unknown.

The dysregulation of the EV/EC system, which might be triggered by iron overload, is likely to represent one of the molecular events underlying the development of beta thalassemia major (TM)-induced OP. As demonstrated in postmenopausal OP, the osteoclast hyperactivity in TM patients is strongly associated with a dysregulation of EV/EC receptor expression: TRPV1 and CB1 are upregulated, whereas the protective CB2 receptor is downregulated. The TRPV1 pharmacological desensitization causes the increase in CB2 receptor expression. Therefore, TRPV1 stimulation does not alter TRAP levels, demonstrating that the channel activation and desensitization are iron dependent [77]. Finally, considering the central role of the EV/EC system in the regulation of bone formation and resorption balance, its pharmacological modulation could revert all the pro-osteoporotic effects induced by estrogen withdrawal in menopause, by glucocorticoids in glucocorticoid-induced OP, and by iron overload in thalassemia major. 

Bone is also commonly affected in cancer. Cancer-induced bone disease can result from the primary disease itself or from therapies, such as adjuvant chemotherapy, administered to treat the primary condition. In the former case, bone loss can be related to tumor-produced circulating hormones and cytokines that compromise the local bone formation [78]. Moreover, OP and fractures are frequently present in long-term cancer survivors [79], therefore early identification and treatment of OP in cancer patients could prevent bone fractures, thereby improving quality of life. 

## 3. Endocannabinoid/Endovanilloid System in Bone Cancer

The EC/EV system has a crucial role in many physiological processes as well as in pathological conditions [12,80,81], such as inflammation, analgesia, immunoregulation, and also in cancer. Munson et al. (1975) and Carchman et al. (1976) observed for the first time a reduction in lung adenocarcinoma cell growth, both in vivo and in vitro, after the administration of D^9^-tetrahydrocannabinol (D^9^-THC), pointing out the anti-proliferative properties of this system [82,83]. D^9^-THC is a metabolite, specifically a phytocannabinoid, produced by *Cannabis sativa* and acting as agonist at CB1 and CB2 receptor level, even though its observed effects are principally mediated by the first one [84]. Although the biochemical mechanisms underlying the anticancer capacities require further investigation, it is already known that the activation of EC receptors induces the synthesis of ceramides, lipids present in the cellular membranes, whose production activates the MAPK signaling cascade and leads to consequent apoptosis and cell cycle arrest [85,86,87,88]. In addition, the activation of TRPV1 receptor with a classical agonist, such as capsaicin, can induce cell death through the increase in intracellular H_2_O_2_ and Ca^2+^ that leads, for example, to a depolarization of mitochondrial membrane [89,90]. Tumor cell death can occur by apoptosis or by necrosis, depending on the cellular context [91]. In osteosarcoma-like G292 cells, capsaicin causes apoptosis [89] as well as in breast cancer cells when it is used both alone and in combination with other modulators (i.e., MRS1477) and the chemotherapy cisplatin [92]. 

The most frequent primary cancers affecting bone are chondrosarcoma and osteosarcoma (OS) [93]. In particular, OS is the most common bone tumor in children and adolescents that preferentially affects areas of active growth [94] and is characterized by pain, limited movement, and high rate of metastasis [95], the majority of which occurs in the lung. In physiological conditions, the bone homeostasis is maintained by a balance between osteoclast-mediated bone resorption and osteoblast-enhanced bone formation. In bone tumors, including OS, this balance is disrupted [94]. Our knowledge about bone malignancies derives from in vitro studies and from in vivo studies on animal models, among which *Danio rerio* (zebrafish), which is a reliable model of human bone tumor [80,96]. In the literature, several studies report a connection between inflammatory status and tumor progression [97,98]. CB1 is known to promote inflammation [99], whereas CB2 regulates the magnitude of the inflammation as observed in the neutrophils isolated from pro-inflammatory phenotype *CB2-/- mice* that exhibit an enhanced migration and adhesive properties. These cell features were inhibited once treated with a CB2 agonist [100]. TRPV1 reduces the release of pro-inflammatory cytokines, such as TNF and IL-6, when stimulated with specific agonists (i.e., capsaicin and RTX) [101]. Hence, considering the influence of the EC/EV system on inflammation and the connection between inflammation and tumor, it is worth investigating the therapeutic potential of the EC/EV system in tumor. In 2017, it was demonstrated that CB2 and TRPV1 receptors can interfere with tumor growth and invasion in several OS cell lines (MG-63, U-2 OS, MNNG/HOS, Saos-2, KHOS/NP, and Hs888Lu) when stimulated, respectively, with JWH-133 and RTX, two selective agonists [102,103]. This is strong evidence of the EC/EV system potential as therapeutic target in OS. The same research group confirmed the possibility of using this system for OS management not only directly triggering the EC/EV system, but also using it as co-adjuvant of a proteasome inhibitor already used as an anticancer drug in other malignancies [104]. Indeed, they observed a synergic anticancer effect when bortezomib (BTZ) is used together with selective agonists on the EC/EV system (JWH-133 and RTX) in the HOS cell line, where both a BTZ-mediated inhibition of proteasome and an activation of CB2 and TRPV1 receptors are induced. An increase in apoptotic cell percentage, cell cycle arrest, and reduction in cell migration in in vitro experiments were observed. In the same year, Roy et al. highlighted the importance of a correct dietary regimen in OS therapy, in particular of omega-3 fatty acids, such as docosahexaenoic acid (DHA) [97]. It is enzymatically converted into docosahexaenoylethanolamide (DHEA), which is an endocannabinoid that suppresses tumor proliferation, migration, and also the angiogenic process in a murine model of OS. This effect seems to be mediated from CB1 receptor in a manner that must be deeply investigated. Another CB1-mediated effect was observed last year by Hsu et al. when they treated MG-63 cell line with anandamide, an important endocannabinoid acting as partial agonist of CB1 receptor and also as antagonist of TRPV1 [105]. They observed an increase in intracellular calcium in the cited OS cell line with consequent phosphorylation of p38 MAPKs (mitogen-activated protein kinases) and activation of the apoptosis effector protein caspase-3. As we stated previously, CB2 receptor is also involved in tumor processes. Osteoclasts from OS patients have reduced levels of this receptor compared with healthy subjects, and this condition is even more marked when patients are undergoing chemotherapy (Figure 3A,C). The immediate consequence is a decrease in bone mass density that could lead to osteoporosis over the years. Bellini et al. demonstrated that the use of mifamurtide, during canonical chemotherapy, can improve this condition, counteracting the hyperactivity of OS osteoclasts and thus reducing the probability of developing osteoporosis in OS patients [106]. Indeed, they observed that mifamurtide leads to a reduction of pro-osteoporotic markers (TRAP, PKCβ2, and TRPV1) and to an increase in CB2 levels in healthy osteoclasts, whereas in osteoclasts from OS patients, mifamurtide is able to revert the chemotherapy-induced effects on bone resorption. Patients with primary bone cancer, like OS, and bone metastases commonly acquire cancer-induced bone disease (CIBD), presenting several troubles such as bone pain, restricted mobility, high rate of fractures, nerve compression, and hypercalcemia. It has been observed that pharmacological and genetic manipulation of CB2 reduces the progression of CIBD [107]. In particular, its activation with specific ligands acts directly on tumor-inducing apoptosis or necrosis in malignant cells, but also on osteoclasts, inhibiting their formation and differentiation. CB2 activation is also associated with a reduction in tumor angiogenesis, maybe due to an autocrine inhibition in vascular endothelial growth factor (VEGF) production by the tumor itself. In 2013, Lozano-Ondoua et al. were the first ones who, using the CB2-selective agonist JWH015, observed a reduction in the number of breast cancer cell line 66.1 in the intramedullary cavity of mice after an injection that mimicked a bone metastasis condition [108].

## 4. EC/EV System in Bone Cancer Pain

The EC/EV system also plays a role in modulating bone cancer pain in mice models [109]. The analgesic function of CB1 receptor is well known in the scientific community, however its stimulation induces psychoactive effects which limit the therapeutic application of CB1R ligands [81]. The discovery of the analgesic potential of CB2 stimulation is instead more recent [110,111,112]. In 2010, Curto-Reyes et al. demonstrated that the selective stimulation of CB2 receptor, by means of AM1241, a CB2-selective agonist extensively used in pre-clinical studies, inhibited tumor-derived pain in mice inoculated with NCTC 2472 osteosarcoma cell line [113]. This is strong evidence of CB2’s role in alleviating pain in OS animal models. The same year, Lozano-Ondoua et al. demonstrated that the administration of AM1241 in a murine model of osteolytic sarcoma inhibited bone cancer-induced pain as well as attenuated cancer-induced bone degradation, suggesting a single therapy for cancer-induced bone pain, bone loss, and bone fractures without inducing central nervous system side effects [114]. Considering that opiates, widely used to trait this type of pain, can enhance bone destruction, CB2 agonists could be a valid alternative. In addition, TRPV1 modulation is responsible for bone cancer pain alleviation, as Bao et al. demonstrated in 2015 in a bone cancer animal model (Wistar rat) [115]. Indeed, its expression levels are highly increased in bone cancer within dorsal root ganglion (DRG) neurons [116]. The topical application of Xiaozheng Zhitong Paste (XZP) causes a decrease in expression levels of EV receptor and the inhibition of PAR2 pathway, thus obtaining a reduction in cancer-induced pain. Moreover, Niiyama et al. obtained a similar result after the i.p. administration of I-RTX, a potent TRPV1 antagonist, in mice in whose femur they first implanted osteosarcoma cells [116].

## 5. Conclusions

Bone mass loss due to several reasons is still a striking health problem in today’s society. 

Considering the involvement of EC/EV receptors in bone remodeling, one is likely to hypothesize that their pharmacological modulation could restore bone balance in all pathological conditions where an altered osteoblast/osteoclast activity is observed. The modulation of the EC/EV system seems to revert all the pro-osteoporotic effects induced by estrogen withdrawal in menopause, by glucocorticoids in glucocorticoid-induced OP, and by iron overload in thalassemia major.

Moreover, based on evidence from the most recent studies (e.g., CB2 and TRPV1 agonists have anti-proliferative, pro-apoptotic, and anti-invasive effects in the most frequent primary cancer affecting bone, namely, osteosarcoma), the EC/EV system could also be a very promising therapeutic target in primary bone tumor and in chemotherapy-induced osteoporosis. 

Furthermore, targeting TRPV1 receptors could also be beneficial in reducing cancer pain, due to the well-known analgesic properties of TRPV1 stimulation in animal models with bone cancer pain. Therefore, pharmaceutical industries should consider investing in hybrid molecules able to stimulate CB2 receptors and, at the same time, desensitize or antagonize TRPV1 channels. These molecules are not psychoactive and therefore constitute a good therapeutic candidate in transitional studies that aim to validate the clinical use of CB2 and TRPV1 agonists in osteoporosis treatment.

## Figures and Tables

**Figure 1 ijms-20-01919-f001:**
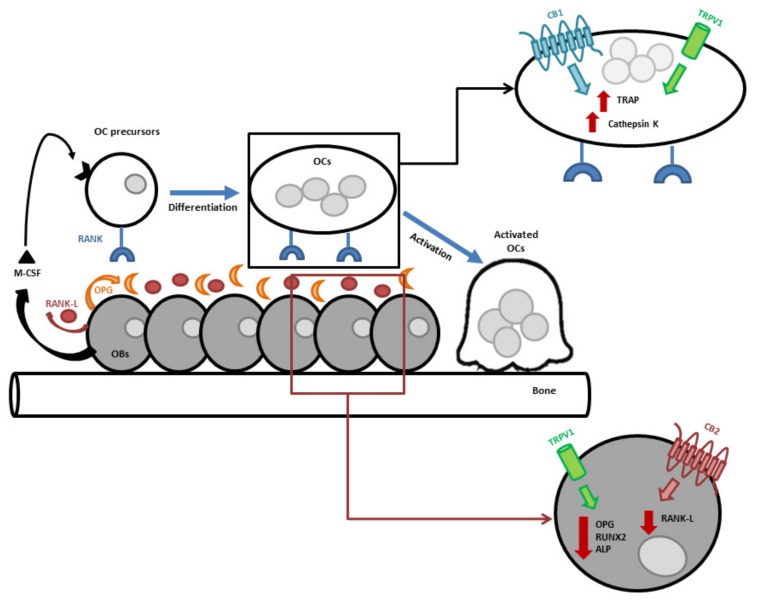
Signaling molecules involved in bone mass maintenance. The cannabinoid receptor type 1 (CB1), the cannabinoid receptor type 2 (CB2), and the transient receptor potential cation channel subfamily V member 1 (TRPV1) contribute to the maintenance of bone mass. CB2 receptor acts as inductor of bone matrix deposition, whereas the TRPV1 and CB1 receptors act as inhibitor of osteogenic signaling. M-CSF, Macrophage colony stimulating factor.

**Figure 2 ijms-20-01919-f002:**
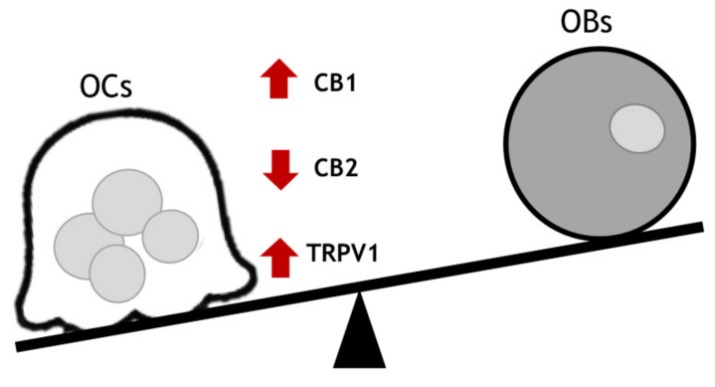
Altered homeostasis in osteoporosis. Activated osteoclasts (OCs) in patients with osteoporosis express low levels of CB2 receptor and high levels of CB1 and TRPV1. CB1 and TRPV1 stimulation activates osteoclasts, while CB2 represents the counterpart for bone mineralization and remodeling via osteoclast inhibition.

**Figure 3 ijms-20-01919-f003:**
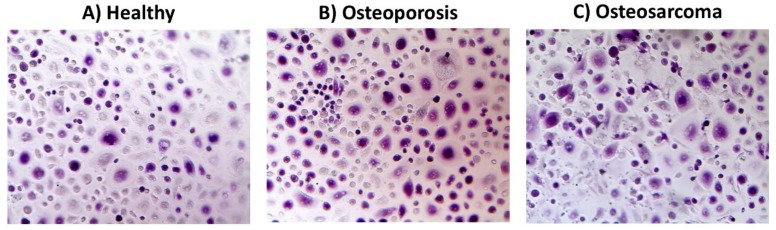
Representation of the multinucleated tartrate-resistant acid phosphatase (TRAP) (+) osteoclasts obtained by colorimetric assay (TRAP assay). The figure shows that osteoclasts (in purple) derived from (**B**) a postmenopausal woman with osteoporosis and from (**C**) an osteosarcoma patient are more active, as shown by the more intense staining and the larger size, with respect to osteoclasts derived from (**A**) a healthy donor.

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
