# Peer review of "The Endocannabinoid/Endovanilloid System in Bone: From Osteoporosis to Osteosarcoma"

_ijms, 2019, doi:10.3390/ijms20081919_

Round 1
Reviewer 1 Report
Chapter 1. Authors provide a quite comprehensive overview of bone physiology and of the EC system.
The role of the EC system in bone should be better described in this section. A figure presenting such mechanisms might be helpful.
Chapter 2. The description of OP is too long. this part should be shorten and integrated with chapter 3.
Chapter 4. The part concerning the treatment of bone pain should be discussed separately.
Author Response
Response to Reviewer 1 Comments
We thank the reviewer for his/her comments and for the opportunity to improve our manuscript.
Point 1: Chapter 1. Authors provide a quite comprehensive overview of bone physiology and of the EC system. The role of the EC system in bone should be better described in this section. A figure presenting such mechanisms might be helpful.
Response 1: As suggested, in the revised version of our Review, we better describe the role of EC system in bone (lines 61-87) and in order to make the described mechanism clearer we made a new Figure (Figure 1).
Point 2: Chapter 2. The description of OP is too long. this part should be shorten and integrated with chapter 3.
Response 2: As required, chapter 2 (the description of Osteoporosis) was shortened and integrated in Chapter 3.
Point 3: Chapter 4. The part concerning the treatment of bone pain should be discussed separately.
Response 3: As required, in the revised version of our Review, the part concerning the treatment of bone pain is separately discussed
Reviewer 2 Report
The manuscript « Endocannabinoid/endovanilloid System in bone: from osteoporosis to osteosarcoma » presents a review of the most recent research reports describing the identification and characterization of cannabinoid receptors (CB1, CB2) and endovanilloid receptor (TRPV1) on bone cells. In this review, main results of the selected research reports are well-synthetized and presented in the context of osteoporosis and bone cancers. Those receptors influence both osteoclast and osteoblast differentiation as shown by inhibition or deletion assays and then, they may be a therapeutic target to revert osteoporosis. In the context of osteosarcoma, receptors CB1, CB2 or TRPV1 influence viability and drug-sensitivity of tumor cells but also modulate inflammation and angiogenesis within the tumor microenvironment. This review is of high interest and is well organized.
Few modifications should be done:
- In the first paragraph, from lines 31 to 45, all references have to be corrected.
- Line 49, please change “phisiologically” by “physiologically”.
- All gene symbols should be indicated in italic front, following the recommendations for gene and protein nomenclature. i.e. line 136 change “CNR2 gene” by “CNR2 gene”, line 178 change “TRPV1-/- mice” by “TRPV1-/- mice”.
- A higher magnification should be used for pictures on figure 2.
- The red arrow on figure 2A does not indicate a TRAP+ multinucleated cell, while one is clearly observed in the middle of the same picture.
- Please change the end of the sentence line 237 “whose production activates the ERK signaling cascade” by “whose production activates the MAPK signaling cascade”. Indeed within the MAPK family, not ERK but p38 MAPK is implicated in cell death induced by ceramide, as described by Niaudet et al. (Cell Signal. 2017; doi: 10.1016/j.cellsig.2017.02.001).
Author Response
Response to Reviewer 2 Comments
We thank the reviewer for his/her comments and for the opportunity to improve our manuscript.
Point 1. In the first paragraph, from lines 31 to 45, all references have to be corrected
Response 1: The References indicated have been corrected.
Point 2. Line 49, please change “phisiologically” by “physiologically”.
Response 2: As suggested, we corrected the mistake.
Point 3. All gene symbols should be indicated in italic front, following the recommendations for gene and protein nomenclature. i.e. line 136 change “CNR2 gene” by “CNR2 gene”, line 178 change “TRPV1-/- mice” by “TRPV1-/- mice”.
Response 3: As suggested, in the revised version of our Review, we indicated in italic front all gene symbols.
Point 4. A higher magnification should be used for pictures on figure 2.
Response 4: We aimed to show the different number of active Osteoclasts in a wide field. If we zoom in the picture our aim would not be fulfilled anymore.
Point 5. The red arrow on figure 2A does not indicate a TRAP+ multinucleated cell, while one is clearly observed in the middle of the same picture.
Response 5: The red arrow on figure 2A indicated a quiescent osteoclast of a healthy subject in order to bring the differences with osteoporotic and osteosarcoma patients. We realize it could be misleading so we deleted all the arrows to avoid misunderstandings.
Point 6. Please change the end of the sentence line 237 “whose production activates the ERK signaling cascade” by “whose production activates the MAPK signaling cascade”. Indeed, within the MAPK family, not ERK but p38 MAPK is implicated in cell death induced by ceramide, as described by Niaudet et al. (Cell Signal. 2017; doi: 10.1016/j.cellsig.2017.02.001).
Response 6: As suggested, we changed it in the revised version of the manuscript.